# Effectiveness of Respiratory Rehabilitation in COVID-19’s Post-Acute Phase: A Systematic Review

**DOI:** 10.3390/healthcare11081071

**Published:** 2023-04-08

**Authors:** Matteo Tamburlani, Rossana Cuscito, Annamaria Servadio, Giovanni Galeoto

**Affiliations:** 1Local Health Board Roma 2, 00145 Rome, Italy; 2Master’s Degree Course in Rehabilitation Sciences of the Health Professions, University of Rome Tor Vergata, 00133 Roma, Italy; 3Departement of Human Neurosciences, Sapienza University of Rome, 00185 Rome, Italy; 4Neuromed, IRCCS, 86077 Pozzilli, Italy

**Keywords:** COVID-19, Coronavirus, post-acute, pulmonary rehabilitation, pulmonary function, respiratory physiotherapy, telerehabilitation

## Abstract

Background: The COVID-19 pandemic, caused by the new grave and acute respiratory syndrome Coronavirus-2 (SARS-CoV-2), generated an unprecedented danger to public health. This condition may impact survivors’ quality of life and includes extensive pulmonary and respiratory outcomes. Respiratory rehabilitation is known for its effects in improving dyspnea, alleviating anxiety and depression, reducing complications, preventing and ameliorating dysfunctions, reducing morbidity, preserving functions and improving subjects’ quality of life. For this reason, respiratory rehabilitation may be recommended for this category of patients. Objective: Our objective was to evaluate the effectiveness and benefits produced by the adoption of pulmonary rehabilitation (PR) programs in COVID-19’s post-acute phase. Material and Methods: A search of relevant publications was conducted using the following electronic databases: PubMed, Scopus, PEDro, and Cochrane Library. A single reviser selected pertinent articles that studied the effects of pulmonary rehabilitation during COVID-19’s post-acute phase in improving the respiratory function, physical performance, autonomy and quality of life (QoL). Results: After an initial selection, 18 studies were included in this systematic review, of which 14 concern respiratory rehabilitation delivered in conventional form and 4 concern respiratory rehabilitation provided in telehealth. Conclusions: Pulmonary rehabilitation combining different types of training—breathing, aerobic, fitness and strength—and not bypassing the neuropsychological aspects revealed itself to be capable of improving pulmonary and muscular functions, general health and quality of life in post-acute COVID-19 patients, besides increasing workout capacity and muscle strength, improving fatigue states and reducing anxiety and depression.

## 1. Introduction

The COVID-19 pandemic, caused by the new grave and acute respiratory syndrome Coronavirus-2 (SARS-CoV-2), identified for the first time in December 2019, generated an unprecedented danger to public health and still represents an extraordinarily impactful event that continues to negatively affect people’s health across the globe. Since the beginning of the pandemic, 640,395,651 confirmed cases worldwide have been registered and 6,618,579 deaths have been recorded so far [1], even though the number of new infections has significantly decreased in the past few months thanks to the ample vaccine campaign conducted by all countries globally. The infection’s clinical spectrum is wide and varied, ranging from asymptomatic infection to a slight sickness of the higher respiratory tract, a moderate illness to a critical one, which can be described as a serious viral pneumonia with respiratory distress, septic shock and/or multiple organ failure [2]. A total of 41.8% of infected subjects developed ARDS and 52.4% of these died [3]. The American-European Consensus Conference on ARDS defined the acute respiratory distress syndrome (ARDS) as a process of nonhydrostatic pulmonary edema and hypoxemia associated with a variety of etiologies and carries a high morbidity and mortality (10 to 90%) [4].

A considerable pulmonary concern has characterized this condition; indeed, only 25.8% of patients had lesions affecting a single lung, whereas 75.7% of patients had lesions affecting both lungs bilaterally [5]. Respiratory damage plays a crucial role within patients who have passed COVID-19, since the removal of the cause of lung damage does not hinder the development of fibrotic and progressive interstitial lung disease. Pulmonary fibrosis is indeed known to be consequence of ARDS [6]. Not surprisingly, reduced diffusion capacity, restrictive pulmonary physiology, ground glass opacity and fibrotic imaging changes were found at the follow-up of COVID-19 survivors [7].

The issue for the survivors of COVID-19 does not terminate with the end of the pulmonary inflammation, since a significant number of patients continue signaling persistent symptoms way beyond the acute phase of the sickness. According to the Office for National Statistics, one person out of five found positive to COVID-19 shows symptoms for a period of 5 weeks or more, while one person out of ten develops symptoms lasting 12 weeks or longer [8].

These medium- and long-term effects—known as post-COVID-19 syndrome, signs and symptoms that continue for more than 12 weeks, or “Long COVID”, including both ongoing symptomatic COVID-19 (from 4 to 12 weeks) and post-COVID-19 syndrome (12 weeks or more)—are extremely varied and extensive [9].

Within 6 months of COVID-19 infection, fatigue and muscular weakness (63%), sleeping difficulties (26%) and anxiety and depression (23%) are the most common symptoms [10].

Data from the UK National Statistical Office suggest that Post-COVID-19 or “Long COVID” syndrome has an incidence rate of 13.7% [11]. This makes rehabilitation measures for the promotion of physical recovery a crucial necessity. Rehabilitation programs play a crucial role in combating the pandemic, in addition to the use of vaccines [12], as they are an effective means of containing the adverse effects of COVID-19 on public health [13].

The National Institute for Health and Care Excellence (NICE) recommends that gradual rehabilitation programs be used within the first 30 days (post-acute phase) to have a maximum impact on recovery [14]. In the literature, there is no unique and officially recognized definition of the post-acute phase; some authors use this expression to indicate the immediate phase after the acute one (after 4 weeks) with persistent symptoms [15].

It thus seems necessary to prepare a multidiscipline and holistic rehabilitation program that considers respiratory rehabilitation tailored on the needs of the single individual to favor complete recovery. Although respiratory rehabilitation was primarily planned for treating chronic lung diseases, numerous reports, guidelines and expert opinions focus on the recommendation of pulmonary rehabilitation in patients recovering from SARS-CoV-2 infection [16,17].

Respiratory rehabilitation aims to ameliorate dyspnea, alleviate anxiety and depression, reduce complications, prevent and ameliorate dysfunctions, reduce morbidity, preserve functions and improve subjects’ quality of life as much as possible [16].

The objective of this study is to conduct a systematic review of the scientific literature to assess the efficacy and benefits of pulmonary rehabilitation (PR) programs in the post-acute phase of COVID-19, which is useful in promoting an improvement in the respiratory functions, autonomy and quality of life (QoL) of people affected by COVID-19 and reduce the incidence and severity of lung complications.

## 2. Materials and Methods

### 2.1. Research

This systematic review was conducted following the international guidelines of the Preferred Reporting Items for Systematic Reviews and Meta-Analyses (PRISMA).

A search of relevant publications was conducted with the use of the following electronic databases: Medline (via PubMed), Scopus, PEDro and the Cochrane Library and was carried out between December 2020 and September 2022.

The literature presented has been vetted through the formulation of a search string common to the following databases: Medline, Scopus, PEDro and Cochrane Library containing keywords and Boolean operators AND/OR in combination as follows: (Pulmonary rehabilitation OR post-acute) AND COVID-19.

On the PEDro database, the search was carried out using the following word couples: Pulmonary rehabilitation AND COVID, COVID-19 AND Rehabilitation, COVID-19 AND Physiotherapy, SARS-CoV-2 AND Rehabilitation, SARS-CoV-2 AND Physiotherapy, COVID-19, SARS-CoV-2.

The search terms are shown in Table A1 in the Appendix A. 

### 2.2. Eligibility Criteria

In alignment with the PRISMA guidelines, inclusion and exclusion criteria were laid out through the definition of the PICO strategy (population, intervention, comparison and outcome), as reported below:-P (population): patients diagnosed with COVID-19 in the post-acute phase and clinically stable were included. Instead, severe COVID-19 cases or acute-phase cases with clinical instability were excluded.-I (intervention): respiratory physiotherapy in its different means were included, either delivered in conventional form (in person) or through telemedicine. Other forms of rehabilitation were excluded.-C (comparison): patients who only receive standard assistance/cure or receive no cure.-O (outcome): improvement of respiratory function and physical performance, reduction in dyspnea and fatigue and improvement of autonomy and quality of life in patients affected by COVID-19. The Table 1 shows the PICO strategy used. 

In addition, the research was limited to the use of the following filters:-Publication date (year) of the articles: we have included in the revision those articles that had been published in multimedia databases in the time range spanning from the 1 January 2020 to the 27 September 2022.-Language of publication of the articles: all studies that were not redacted in either Italian or English were excluded.-Type of study: in the present review, we included randomized controlled trials, cohort studies, declarations of consent and practical guidelines on pulmonary rehabilitation for SARS-CoV-2.

### 2.3. Selection of the Studies

After the deletion of duplicates using the EndNOTE software 20, an editor evaluated the studies taken from the databases based on the title and later the studies’ abstract. After this initial selection, we moved on to analyzing the studies’ full texts to determine whether they met the required criteria of inclusion/exclusion. Following the full-text analysis, a decision was made as to which articles must be included in the final review. For those articles over which there were some doubts, the decision was taken after another proofreading of the full text.

### 2.4. Data Collection Process

After inclusion, the studies’ characteristics, aims and results were extracted and summarized using an extraction table. More specifically, the following data were gathered: name of the first author, publishing date, title of the article, study’s design, size and characteristics of the sample (sex, average age of patients), rehabilitation protocol, frequency of intervention, primary and secondary outcome measures, evaluation time and obtained results.

### 2.5. Evaluation of Methodologic Quality

The randomized controlled trials (RCT) included were evaluated using the Physiotherapy Evidence-Based Database (PEDro) scale, considered a reliable instrument allowing critical evaluation of methodologic quality in experimental studies on physical therapy. The final score of the PEDro scale varies from 0 to 10 with each satisfied element contributing 1 point. No score is assigned if one criterion is not described or is unclear.

The Newcastle—Ottawa quality assessment scale evaluated the cohort and observational studies. The NOS scale allows the evaluation of the quality of non-randomized studies (cohort studies and case–control studies) with its design, content and ease of usage allowing for the incorporation of quality evaluation in the interpretation of meta-analytic results. A “star system” was developed in which a study is evaluated based on three outlooks: the quality of the selected cohort, the comparability of cohorts and the obtained results, thus assigning up to a maximum of nine stars.

The Newcastle-Ottawa Scale quality instrument is scored by awarding a point for each answer that is marked with a star symbol ♦. Possible total points are 4 points for Selection, 2 points for Comparability, and 3 points for Outcomes with a maximum of 9 points.

The practical guidelines and declaration of consent were included, as they are relevant for the revision, but not methodologically evaluated.

## 3. Results

### 3.1. Selection of Studies

The total number of articles identified through multimedia database research was 3542, of which 692 articles were duplicates and thus excluded. A total of 2423 articles were also excluded based on their title since it did not fit the required inclusion criteria; subsequently, 320 articles were excluded after their abstract was read and another 87 articles were eliminated after their full text was examined. In the end, 18 articles were deemed useful and relevant: 4 randomized controlled trials (RCT), 7 cohort studies, 5 observational studies, 1 practical guideline and 1 declaration of consent.

The diagram in Figure 1 shows the scheme that was followed to select the articles.

### 3.2. Evaluation of the Methodologic Quality

The evaluation of the randomized controlled trials was carried out with the aid of the PEDro scale. Following the PEDro criteria, the studies’ quality can be sorted in low quality (0–3 points), medium quality (4–7 points) and high quality (8–10 points), with 10 being the highest score. A study with a score of at least 6–10 points is of medium-high quality. The final score obtained in the evaluation of the randomized controlled trials included varies from 5 to 7; with an average of 6.3, the included RCTs are of medium-high quality [18,19,20], and only one RCT is of medium quality [21]. The evaluation of the randomized controlled trials is shown in Table 2.

The cohort and observational studies were evaluated using the Newcastle—Ottawa scale. The final score obtained in the evaluation of said studies varies from a minimum of 3 to a maximum of 7 points, with an average of 5.1. The 12 evaluated studies [22,23,24,25,26,27,28,29,30,31,32,33] were of heterogeneous quality with generally low scores. A total of five cohort studies [25,29,30,31,32] and two observational studies [24,33] obtained less than 6 points on the NOS scale and were categorized as being of low methodological quality. In this case, the results of these studies should be considered as questionable compared to the others. This significant difference is attributable to certain issues such as the absence of control groups, the application of a rehabilitation protocol on a small cohort made up of a low number of patients and, lastly, a lack of adequate follow-up, which is necessary to better understand the long-term benefits of the variables under scrutiny. The evaluation of the cohort and observational studies is shown in Table 3 and Table 4.

### 3.3. Features of Studies

The characteristics of the studies included are given below in the way the data were extracted.

### 3.4. Population

The analyzed studies included an overall population of 1592 subjects affected by COVID-19 in the post-acute phase. The number of patients for studies varies from 23 [19] to 518 [32]. The average age of patients varied from 49.17 [21] to 72.15 years [25]. The average age of the overall population was 58.12 years. Most patients were male. The data related to the overall population included in the analyzed studies are shown in Table 5.

### 3.5. Intervention

The different approaches used within the selected studies include respiratory rehabilitation delivered through different protocols and with a varying number of sessions and methods of intervention. The approaches were different for all studies and w described for each one.

Four studies adopted a respiratory physiotherapy program comprising deep breathing, inhaling muscle training and bronchial hygiene techniques [19,23,29,33]. One study kept to the European Respiratory Society and American Thoracic Society’s recommendations (ATS/ERS) for that concerning breathing exercises [22].

Another study provided a program of respiratory rehabilitation based on thoracic clapping and vibrations, breathing control and thorax–abdominal coordination exercises, breathing exercises using a pep bottle, forced inhaling/exhaling and the use of incentive spirometers [24]. Three studies envisioned the teaching of breathing control (breathing with pursed lips, secretion mobilization and diaphragmatic breathing) and controlled coughing exercises [20,30,32]. One study combined muscular training with active cycle breathing training (ACBT) [26,31]. Instead, two studies gave a less detailed definition of the respiratory rehabilitation program [25,27].

Four studies envisioned a telerehabilitation program that provided for a virtual and supervised respiratory and muscular rehabilitation program, deployed through a telemedicine platform [18,21,28,31]. Two of these studies (Hameed et al. and Pehlivan et al.) describe a rehabilitation program delivered on a telerehabilitation platform. In Hameed et al. [28], diaphragmatic breathing exercises, spirometry incentive, sit-to-stand, standing gear, shoulder scaption, standing heel lifts, sidestepping and wall flexion are proposed. In Pehlivan et al. [20], patient education, rhythm run/autonomous walk in the corridor, breathing exercises and an active cycle of breathing techniques is proposed, in addition to range of motion exercises and standing squats. Instead, in two of these studies, telerehabilitation program is proposed through a smartphone app. In Capin et al. [18], the program includes breathing and compensation techniques, high-intensity strength training, aerobic and cardiovascular exercises, exercise balance, functional activities, stretching, and lifestyle coaching and motivational talk. In addition, this application is used for facilitating self-directed intervention outside supervised sessions. In Li et al. [19], the use of an application for smartphones called “RehabApp” is described, and this study includes chest breathing and expansion and aerobic exercise in a three-level exercise plan with the difficulty and intensity programmed to increase over time.

All of the studies adopted a rehabilitation protocol in which respiratory physiotherapy is deployed with strength and resistance training of both the upper and lower limbs, balance and coordination exercises, aerobic exercise based on walking/treadmill and cycling and muscular relaxation techniques. Five studies implemented educational sessions, in conjunction with the rehabilitation protocol, which concerned dyspnea, cough, fatigue, fear and anxiety, memory and concentration, self-management of ordinary activities and back-to-office activities [25,27,29,30,32]. Only four studies included psychosocial support with lifestyle and motivational coaching, nutritional consultation and occupational therapy as part and parcel of the rehabilitation protocol [18,19,23,29].

Concerning the length of the interventions, treatment length varies from study to study, from a minimum of one session [28] to a maximum of five–six sessions per week [32], distributed on a time interval spanning a minimum of 3 weeks [27] to a maximum of 12 weeks of treatment [26]. The intensity of the rehabilitation action is also quite heterogeneous and comprises between 20 [32] and 60 min per session [28] (Table 5).

### 3.6. Comparison

Not all the studies included had a control group. One of the studies compares three experimentation groups, each undergoing virtual physical therapy by means of a telemedicine platform, home-delivered physical therapy, an autonomous physical exercise program and one control group that receives no treatment [28].

Three of the included studies made a comparison between cohorts undergoing the same rehabilitation program, which differ in the type of COVID-19 they show: mild/moderate and severe/critical COVID-19 [27], acute/grave/serious/mild COVID-19 [29] and patients on ventilation or not [30].

Three studies compare two cohorts undergoing the same rehabilitation program but draw a distinction between: COVID-19 and non-COVID-19 patients [22], COVID-19 patients and patients with other kinds of pneumonia [23], COVID-19 patients and patients with other types of lung disease [32].

Three studies included a control group which received educational sessions [18,21,31]. Instead, two studies included a control group that received no treatment [20,26].

Lastly, four included studies did not exhibit any control group [19,24,25,32] (Table 5).

### 3.7. Outcome

The analyzed studies show the main results related to two macro-areas: lung function and functional capacity (exercise), which were evaluated in almost all of the studies using different scales and other tools.

The test which was most used to evaluate functional capacity was the six minute walking test (6MWT) [19,21,22,23,24,26,29,30,31,32,33]; one study used the shortened version of the test, namely, the two minute walking test [28]; one study used a different test, namely, the incremental shuttle walking test and the endurance shuttle walking test (ISWT and ESWT) [25]; and one study used both the six minute walking test (6MWT) and the endurance shuttle walking test (ESWT) [27]. Apart from the six minute walking test (6MWT), two studies also used as a stress test the 30 s sit-to-stand test [26,28], while two studies evaluated the functioning of lower limbs using the short physical performance battery (SPPB) [28,31].

The patients’ physical functional, psychological and social state was mainly evaluated using the functional independence measure (FIM) [20,23,30,32]. Only three studies evaluated autonomy in the activities of daily life (ADL) through the Barthel index (BI) [19,24,33].

Only two of the studies considered did not use the six minute walking test as their functional capacity assessment (6MWT), but rather used the time up and go test (TUG) to evaluate patients’ mobility [18,19].

Functional capacity was considered a primary outcome in 11 studies [19,22,23,25,26,27,28,30,31,32,33] and as a secondary outcome in 3 studies [20,24,29].

Pulmonary function was examined in only seven studies using spirometry and plethysmography, using the following parameters: forced expiratory volume in the 1st second (FEV1), forced vital capacity (FVC), residual volume (VR), maximum and minimum inspiratory pressure (Pi max/min), total lung capacity (TLC) and carbon monoxide alveolar–capillary diffusion (DLCO) [19,20,21,22,29,30,32]. In addition to all parameters for pulmonary function above, three studies also included blood gas analysis [19,29,30]. On the other hand, nine studies did not include the evaluation of pulmonary function as an outcome [18,23,26,27,28,31,33]. The pulmonary function was considered a primary outcome in five studies [19,21,22,30,32] and as a secondary outcome for two studies [21,29].

Dyspnea was considered as an outcome only in seven studies. To assess dyspnea, the widest-spread scale used was the Modified British Medical Research Council Questionnaire (mMRC) [18,19,20,24,27,29]; only one study assessed dyspnea using the Borg scale [22]. Dyspnea was considered a primary outcome in three studies [22,29,31] and a secondary outcome in four studies [18,21,24,27]. Nine of the selected studies did not assess dyspnea [21,23,25,26,28,30,31,32,33].

Within the outcomes that were considered, we also have those related to the quality of life in correlation to health (QoL), which was analyzed in nine studies and evaluated through different scales: St George questionnaire [22,31], chronic respiratory questionnaire [30,32], short form 36 [20,27], short form 12 [21] and Euroquol 5D [25,29].

Anxiety and depression were also evaluated with different scales: the hospital anxiety and depression scale (HADS) [22,25,30,32], the general anxiety disorder-7 (GAD-7) and the patient health questionnaire-9 (PHQ-9) [27,28,29] and the Beck depression inventory (BDI) [31].

In addition to this, we must also add cognitive function [18,25], fatigue [22,25,29], equilibrium [18], clinic fragility [18], pain [31] and subjective change in health status [29]. (Table 5).

### 3.8. Summary of Results

The analysis of the obtained results highlights that, in the post-acute phase of COVID-19, a complete rehabilitation program, which is most frequently adopted, provides for respiratory physiotherapy deployed together with aerobic exercise and both strength and resistance exercise, carried out both in person and remotely through telemedicine platforms.

Indeed, out of the 18 studies considered in this systematic review, 12 provided for a pulmonary rehabilitation program deployed in hospital or outpatient settings, while 4 present a remotely deployed rehabilitation program through telemedicine platforms. To this, a practical guideline and a declaration of consent added, which gives suggestions for the planning of rehabilitation programs for COVID-19 patients in different phases.

Specifically for the post-acute phase, the following actions are suggested: disability recovery, peripheral muscle strengthening, thoracic physiotherapy with respiratory muscle training, dry nonproductive cough management and bronchial clearance techniques with appropriate preventive measures. To this, we must add aerobic exercise < 3.0 MET with progressive intensity increase in patients with no or slight disability (SPPB > 10; Barthel index > 70); a complete pulmonary rehabilitation program is necessary in patients with moderate-to-severe disability (SPPB < 10; Barthel index < 70). During exercise training, the continuous monitoring of essential parameters such as blood pressure, SpO2, dyspnea and perceived effort (Borg scale) is recommended. The use of continuous or temporary positive expiratory pressure devices with or without oscillation (PEP, TPEP and OPEP) is suggested for patients with hypersecretion, either alone or in combination with lung expansion strategies to increase lung volume, better control exhalation and facilitate peripheral and proximal mucus movement [34,35]. Based on the protocols suggested by the studies, we hereby summarize the results obtained for the main outcomes that were examined: In evaluating functional capacity, which is the primary outcome for almost all included studies, we detected improvements in all groups that underwent a complete rehabilitation program; in more detail, three of the studies taken into account detected significant improvements in all groups, which differed for the type of COVID-19 they exhibited: light/moderate COVID-19 and severe/critical COVID-19 undergoing the same rehabilitation program showed a slight difference in favor of the patients with severe/critical COVID-19 [27,29,30].

Three studies noted significantly better improvements in COVID-19 group patients compared to the control group, which received the same rehabilitation program but composed of patients suffering from common pneumonia or other lung diseases [22,23,32].

Four studies analyzed a single cohort of COVID-19 patients without a control group and have noted a significant improvement in functional capacity in terms of an increase in the 6MWT, the ISWT, the ESWT and the SPPB, as well as an improvement in autonomy in daily life activities (ADL), assessed through the Barthel index. [19,24,25,33]. The four studies that comprised a rehabilitation program deployed through telemedicine registered significant improvements not only regarding functional capacity but also in symptoms and quality of life, in favor of the group that received the complete rehabilitation program remotely through a telemedicine platform compared to the control group that received no therapy or brief educational instructions [18,21,28,31].

Lastly, two studies highlighted a considerable improvement in functional capacity including quality of life in favor of the group that received the complete lung rehabilitation program with respect to the control group that received no treatment [20,26].

In evaluating the pulmonary function, another outcome analyzed by most studies considered, the same instruments were used; however, heterogeneous results were obtained. Two of the studies noted statistically significant improvements in pulmonary function in favor of the group that received the complete rehabilitation program concerning the control group which received no therapy [20,21].

Two other studies, which compared two groups constituted by COVID-19 patients and patients affected by other pulmonary disease or respiratory failure, found a sizeable improvement in the lung function of the COVID-19-patient group.

Another study analyzed a single cohort of COVID-19 patients without a control group and registered a considerable improvement in lung function, which is still compromised even after the rehabilitation program [19]. One of the studies traced significant improvements in all groups considered, which underwent the same rehabilitation program. The patients differed for the type of COVID-19 they presented: grave/severe/moderate. The results also show a slightly more favorable outcome for those with grave/severe COVID-19, concerning lung function and quality of life and fatigue [29].

Lastly, only one study did not exhibit any significant differences in pulmonary function between the two groups that underwent the same rehabilitation program but were distinguished between ventilated and unventilated [30].

In the analyzed studies, it was observed that COVID-19 patients underwent a complete respiratory rehabilitation program with breathing, aerobic, strength and endurance exercises, carried out both in hospital and remote with telemedicine, recording a gradual recovery of functional capacity, the operating capacity that translates into an initial increase in the distance traveled to 6MWT. This results in a gradual increase in autonomy and improvement of the quality of life. Some studies have shown that the same rehabilitation program focused on respiratory function could improve functional capacity even in patients with different stages of COVID-19 (Table 5).

## 4. Discussion

Our systematic review was carried out to analyze the role of a multidimensional respiratory rehabilitation program in improving the functional impairments detected in post-acute COVID-19 patients to evaluate the effectiveness and the benefits derived and identify specific protocols that may be included within tailored rehabilitation programs.

The results suggest that a complete rehabilitation protocol comprised of respiratory physiotherapy, aerobic training and strength and resistance training is beneficial to ameliorate the typical fallouts in COVID-19 patients. However, the low number of high-quality studies in the literature does not allow the recognition of an elective protocol. Stemming from the suggestion given in the guidelines, recommendations and declarations of consent [34,35,36] that have followed one another in these two years of the pandemic, we can see that an aspect in common for almost all of the studies is the centrality of physical exercise, which represents the key aspect in treating pulmonary illnesses. This is because it allows for a progressive improvement of peak absorption of lung oxygen, functional capacity, muscular strength and size, systematic oxidative stress and health-correlated quality of life [37].

Therefore, the studies we analyzed in this review show results related to two macro-areas: lung function and exercise capacity, evaluated by the six minute walking test (6MWT).

The element that distinguishes the rehabilitation protocols is the type of exercise they suggest: some protocols consider deep breathing, training of inspiratory muscles and bronchial hygiene techniques with a positive finding in the functional tests used, significant results and performance improvement in 6MWT, Barthel index, functional independence measure and lung function tests such as FEV1, CV, TLC, TLCO [23,29,31,33]. Only one study proposes thoracic clapping and vibration, breathing control exercises and thoracoabdominal coordination exercises, breathing exercises with the pep bottle, forced inhalation/exhalation and the use of an incentive spirometer with a significant improvement in the main functional test used: 6MWT, Barthel index and perceived effort with the Borg scale [24]. Others planned the teaching of breathing control: breathing with wrinkled lips, mobilization of secretions, diaphragmatic breathing and controlled coughing exercises with a significant improvement in functional abilities and therefore positive results in 6MWT, functional independence measure and positive results on lung function test. This results in a general improvement in the quality of life [21,30,32].

Respiratory physiotherapy is associated with strength and endurance training of the upper and lower limbs, aerobic exercise, muscle relaxation techniques and, in one case, balance and coordination exercises [24] to achieve progressive functional improvement and a reduction in the level of disability. These rehabilitation programs have allowed the achievement of a gradual recovery of exercise capacity and extensive improvements in muscle strength, balance, reduction in dyspnea and its impact on daily life activities [22,23,24,29,30,32,33].

However, it must be recognized that respiratory rehabilitation does not limit itself to breathing muscle training, but it is indeed, as stated by the American Thoracic Society (ATS) and The European Respiratory Society (ERS), a global approach based on a deep evaluation of the patient eligible for tailored therapy. This therapy must include, but is not limited to, the re-education of physical effort, educational activities aiming at a change in lifestyle and bad behavioral habits and improving lifestyle [38].

To this end, certain studies envisioned coupling the rehabilitation protocol with educational sessions aided by handouts that included: dyspnea, cough, fatigue, fear and anxiety, memory and concentration, self-handling of daily activities and back-to-office activities [25,27,29,30,32].

Only in a few cases was psychosocial support included as part and parcel of the rehabilitation protocol, together with lifestyle coaching and motivational interview [18,23,29,31], to favor the improvement of physical and psychological health, promote therapeutic adherence and ameliorate health in general in patients affected by respiratory disease. In these cases, there was a progressive improvement in the quality of life and a reduction in the symptoms of anxiety and depression, although with minimal dimensions.

Both the educational and the psychosocial components hold key roles in the rehabilitation program; they indeed amount to an important factor that can influence one’s perception of the functional condition and one’s health status in general. This rehabilitation program can also improve patient compliance and better outcomes in terms of limitations and functionality, as well as rapid return to work and social activities.

A comprehensive and multidimensional respiratory rehabilitation program showed positive outcomes in all examined COVID-19 patients. However, most of the studies in the literature focused more on evaluating the effectiveness of rehabilitation in patients who survived intensive care, and little is known about non-ICU-COVID-19 patients. Some studies report how a comprehensive respiratory rehabilitation program can significantly improve functional outcome measures in patients with severe–critical COVID-19 [27,29]. Only one study reports no significant differences in the functional outcomes analyzed at discharge among patients [23].

The novelty suggested in this review of scientific literature is its focus on remote rehabilitation programs. The studies showed that a remote rehabilitation program deployed through telemedicine and under supervision is safe, feasible and effective. The data obtained reveal how a good efficacy and foremost significant improvements from the functional point of view in patients involved, as shown by the increase in distance walked in the six minute walking test (6MWT), to which we can add the increase in lung function and health correlated quality of life [18,21,28,31].

During the pandemic, there was a high percentage of patients with damage and limitations of physical and lung function, and remote rehabilitation has proved to be a viable alternative that could become a new frontier in rehabilitation, to ensure greater continuity in patient care. Since we have seen the issues healthcare systems experienced in the current pandemic, we consider it more important to assure equity in access to therapy in remote areas, continuity in assistance and support for the management of chronic situations.

Hence, we can state that even though we have the presence in the scientific literature, concerning COVID-19’s post-acute phase and heterogeneous pulmonary rehabilitation protocols in terms of suggested exercises, we can still see that a rehabilitation protocol that combines respiratory physiotherapy, aerobic training and strength, without neglecting the neuropsychological aspect, produces a positive effect on exercise capacity, lung function, dyspnea, fatigue, reduction in anxiety and depression and improvement of quality of life (QoL). This goes to show the fundamental importance of starting complete and tailored rehabilitation protocols in the post-acute phase, especially in those patients that exhibited a grave/critical form of pneumonia, to ensure the obtainment of the highest level of physical, functional, social autonomy possible and improve quality of life in COVID-19 survivors, as well as reducing the incidence of serious lung and functional complications, thus producing benefits from the economic–health point of view.

In carrying out this systematic review, we encountered several issues, the largest of which was the selection of the studies, since the pandemic has hit the world with dramatic effects leaving few possibilities to carry out high-quality clinical studies on a wide range of patients. Hence, most of the current protocols are based on preliminary results, observational studies, cohort studies, experts’ opinions, consent of professionals and previous experiences derived from the acute respiratory distress virus pandemic (SARS). This aspect has determined the vast array of suggested designs proposed by the studies we considered and their different methodologies and respective quality.

Another relevant limit is the absence, for the most part, of a control group, which did not allow the obtaining of absolute proof regarding the efficacy of the rehabilitation protocol proposed for the treatment of COVID-19 derived functional impairments.

Indeed, most of the studies accounted for that suggested the application of a rehabilitation program on a small cohort comprised of a limited number of patients, which further limits the extensibility of the acquired results over the global population. In addition, many studies did not carry out any adequate follow-up, thus needing further verification to better understand the long-term benefits of the suggested protocols for respiratory rehabilitation in the post-acute COVID-19 phase. Finally, a significant limitation is represented by the lack of a unique definition of post-acute phase in the literature, which creates differences between the different studies that have analyzed this phase.

Despite the limitations found, the quality of the studies included is such that it allows the definition of adequate results to the aim of the research for which they were intended.

## 5. Conclusions

To conclude, the consequences stemming from the COVID-19 pandemic caused a sizeable impact on the entire population’s health, producing long-term effects that impair the physical performance and quality of life of COVID-19 survivors. In such a context, pulmonary rehabilitation, in its many facets, holds a primary role by allowing a gradual recovery of lung elasticity and optimal breathing flux, obtaining better alveolar ventilation and increasing oxygenation.

Combining exercises such as aerobic, respiratory, fitness and strength exercises without foregoing the neuropsychological aspect has proven to be apt for ameliorating health status, well-being and quality of life in post-acute COVID-19 patients. It has also increased exercise capacity, improved fatigue levels and inhalator muscle strength and diminished anxiety and depression conditions, which are very frequent aspects that can greatly influence patients’ compliance, the latter being key in successfully implementing of a rehabilitation program.

Therefore, it is advisable to increase the implementation of comprehensive and customized rehabilitation protocols for COVID-19 patients, in which a large part is devoted to respiratory rehabilitation to recover overall function and improve in the quality of life.

## Figures and Tables

**Figure 1 healthcare-11-01071-f001:**
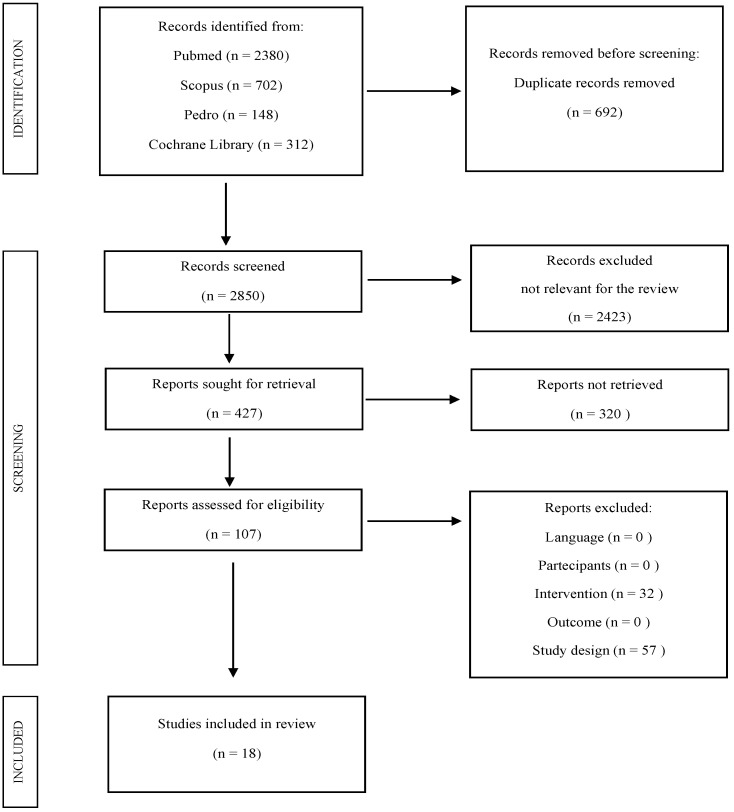
Flow chart.

**Table 1 healthcare-11-01071-t001:** PICO strategy.

P (population)	Patients diagnosed with COVID-19 in the post-acute phase and clinically stable. Severe COVID-19 cases or acute-phase cases with clinical instability were excluded.
I (intervention)	Physiotherapeutic respiratory intervention in its different means, either delivered in conventional form (in person) or through telemedicine.
C (comparison)	Patients who only receive standard assistance/cure or receive no cure whatsoever.
O (outcome)	Improvement of the respiratory function and physical performance, reduction in dyspnea and fatigue and improvement of autonomy and quality of life in patients affected by COVID-19.

**Table 2 healthcare-11-01071-t002:** Pedro scale.

Autor	Item 1	Item 2	Item 3	Item 4	Item 5	Item 6	Item 7	Item 8	Item 9	Item 10	Item 11	Score
Capin 2022 [18]	Y	Y	Y	Y	N	N	Y	Y	N	Y	Y	7
Li 2021 [19]	Y	Y	Y	Y	N	N	Y	Y	N	Y	Y	6
Liu 2020 [21]	Y	Y	N	Y	N	N	N	Y	N	Y	Y	5
Pehlivan 2022 [20]	Y	Y	Y	Y	N	N	N	N	N	Y	Y	6

Y: Yes; N: NO.

**Table 3 healthcare-11-01071-t003:** NOS for cohort studies and prospective observational studies.

Autor	Quality of Selected Cohort	Cohort Comparability	Results Obtained	Score
Al Chikhanie 2021 [22]	♦♦♦♦	♦	♦	6/9
Daynes 2021 [25]	♦♦♦		♦♦	5/9
Dun 2021[26]	♦♦♦♦	♦	♦♦	7/9
Gloeckl 2021 [27]	♦♦♦	♦	♦♦	6/9
Hameed 2021 [28]	♦♦♦♦	♦	♦♦	7/9
Hayden 2021 [29]	♦♦♦	♦	♦	5/9
Hermann 2020 [30]	♦♦♦	♦	♦	5/9
Puchner 2021 [31]	♦♦♦		♦	4/9
Spielmanns 2021 [32]	♦♦♦	♦	♦	5/9

**Table 4 healthcare-11-01071-t004:** NOS for retrospective observational studies.

Autor	Selection	Comparability	Exposure	Score
Busching 2021 [23]	♦♦♦	♦	♦♦	6/9
Curci 2021 [24]	♦♦		♦	3/9
Zampogna 2021 [33]	♦♦		♦	3/9

**Table 5 healthcare-11-01071-t005:** Data Extraction of Selected Studies.

Author Year Title	Study Design	Rehabilitation Protocol	Intervention Frequency	Participants	Outcome Measures	Evaluation Time	Results
Al Chikhanie et al., 2021 [22]	Cohort study	EG and CG: breathing exercises, muscle strengthening, balance and walking when possible, cycling and gymnastics according to current ATS/ERS recommendations	EG: 27.6 ± 14.2 days CG: 29.9 ± 17.3 days	EG: (n.21) -Mean age: 70.9 ± 10.6 -Gender: 14M/7F CG: (n.21) -Mean age: 69.1 ± 9.4 -Gender: 13M/8F	Primary: FEV1; CVF: PImax; PEmax; 6MWT; Tinetti Scale; Borg test; Muscle strength; St George Questionnaire: Quality of Life (QoL); Pichot’s questionnaire; HADS; PCLS Secondary: N/A	At baseline and at the end of the PR program. 6MWT, also performed weekly during PR.	Extensive and rapid recovery of exercise capacity among COVID-19 patients rehabilitated after admission to intensive care, as well as extensive improvements in muscle strength, balance and psychosocial status. Significant improvement in 6MWT, greater in COVID-19 patients (+205 ± 121 m) compared to non-COVID-19 patients (+93 ± 66 m).
Busching et al., 2021 [23]	Retrospective observational study	EG and GC: cardiopulmonary training (cycling, guided walking), strength exercise (free weight, resistance bands), breathing exercises (deep breathing, sputum evacuation), relaxation techniques (progressive muscle relaxation), if indicated psychological, nutritional, speech therapy and occupational therapy.	A minimum of 540 min of patient education and therapy in individual and group settings.	EG: (n.51) -Mean age: 65.8 ± 11.7 -Gender: 38M/13F CG: (n.51) -Mean age: 69.8 ± 9.6, -Gender: 23M/28F	Primary: 6 MWT; FIM: Functional Independence Measure; CRQ: chronic respiratory questionnaire Secondary: N/A	At baseline and at the end of the PR program	Both groups achieved significant improvements in 6MWT, CRQ and FIM. At discharge, COVID-19 patients performed better in 6MWT and FIM, but similar CRQ scores compared to the control group. Regression analysis of the 2 subgroups: COVID-19 intensive care versus non-ICU subgroup: no significant difference in 6MWT at discharge. The outcome of physical functioning in the PR program is similar among critical and severe COVID-19 patients.
Capin et al., 2022 [18]	RCT	EG: breathing and compensation techniques, high-intensity strength training, aerobic and cardiovascular exercises, balance exercises, functional activities, stretching and lifestyle coaching and motivational interviewing. The Health in Motion application used to facilitate self-directed intervention outside of supervised sessions. CG: exercise education with educational handout and weekly check-in phone calls.	12 individual and supervised telerehabilitation sessions provided: 3 times a week in the first week, 2 times a week in weeks 2–4 once a week in weeks 5–6 and 1 single “recall” visit session during week 9 or 10.	EG: (n.29) CG: (n.15) -Mean age: 52 aa -Gender: 23M/21F	Primary: Feasibility through safety and adherence (percentage of sessions attended). Secondary: TUG; MRC; ABC; PROMIS-SF; PHQ8; MoCA	At baseline, 6 weeks after baseline and 12 weeks after baseline (week 12).	The informed multi-component and biobehavioral telerehabilitation program for COVID-19 survivors is safe and feasible. Participants in both groups functionally improved from baseline to 6 weeks and 12 weeks after intervention.
Curci et al., 2021 [24]	Retrospective observational study	EG: Initially: posture changes, passive mobilization, postural drainage, chest clapping and vibration, breathing control exercises and chest-abdomen coordination. Muscle strengthening exercises of the upper and lower limbs, trunk and gluteal muscles. Breathing exercises with the pep bottle, forced inhalation/exhalation and use of the incentive spirometer. Balance and coordination exercises and train yourself to walk for progressive distances. CG: N/A	30 min/set, 2 times a day for the duration of hospitalization. (LOS average in the COVID-19 Rehabilitation Unit was 31.97 ± 9.06 days).	EG: (n.41) -Mean age 72.15 ± 11.07 aa -Gender: 25M/16F CG: N/A	Primary: BI Secondary: MRC; 6-MWT; RPE; Type of respiratory support required; Results of arterial blood gas analysis; Serum levels of laboratory markers.	At hospitalization (T0) and at the end of the PR program (T1).	Statistically significant improvement in the Barthel Index (BI) (84.87 ± 15.56 vs. 43.37 ± 26.00), 6-MWT (303.37 ± 112.18 vs. 240.0 ± 81.31 m) and Borg RPE scale (12.23 ± 2.51 vs. 16.03 ± 2.28). Finally, an improvement also in CT scans in 74.4% of cases
Daynes et al., 2021 [25]	Cohort study	EG: Aerobic exercises based on walking/treadmill, upper and lower limb strength training. Educational sessions with handouts included: dyspnea, cough, fatigue, fear and anxiety, memory and concentration, managing daily activities and returning to work. CG: N/A	2 sessions per week for 6 weeks.	EG: (n.32) -Mean age: 58 aa -Gender: 52%M/48%F CG: N/A	Primary: Incremental running and resistance test (ISWT/ESWT); CAT; FACIT; HADS; EQ5D; MoCA. Secondary: N/A	At baseline and at the end of the PR program	PR produced average improvement within the group of 112 m in the Incremental Shuttle Walking Test (ISWT) and 544 s in the Endurance Shuttle Walking Test (ESWT). The FACIT improved by 5 points, the EQ5D improved by 8 points and the MoCA by 2 points. The CAT score improved by 3 points, while for Anxiety and depression: minimal improvement, equal to 1 point.
Dun et al., 2021 [26]	Retrospective cohort studies	EG: inspiratory muscle training; 30 sets of ruffled lip breathing techniques and active breathing cycle (ACBT); a series of 30 repetitions of maximum voluntary diaphragmatic contractions in the supine position, placing an average weight (1–3 kg) on the anterior abdominal wall to resist diaphragmatic descent; two high-intensity interval workouts of 4 min via bike or treadmill interspersed with 4 min low-intensity intervals. CG: patients that did not perform any PR during the 6 m of convalescence.	3 sessions per week for 12 weeks.	EG (n.27) -Mean age: 54 ± 16 -Gender: 9M/18FCG (n.71) -Mean age: 44 ± 13 -Gender: 36M/35F	Primary: 6-MWT Secondary: -Changes in SARS-CoV-2 specific IgG and IgM immunoglobulins, T lymphocytes and blood chemistry	Baseline, 2 weeks and 6 months.	Patients in the PR group acquired a greater increase in the distance traveled in 6-MWT compared to the control. There were no significant differences between PR and control groups in IgG and IgM specific for SARS-CoV-2, CD3+ T cells, CD8+ cells, CD8+ T cells, CD4+/CD8+ ratio, and all biomarkers.
Gloeckl et al., 2021 [27]	Prospective and observational cohort study	EG1 (COVID-19 mild/moderate) and EG2 (COVID-19 severe/critical): Pulmonary rehabilitation program for COVID-19 patients includes resistance training, strength training, patient education sessions, respiratory physiotherapy, daily life training activities and relaxation techniques. CG: N/A	3 weeks	EG1 (COVID-19 mild/moderate) (n.24) -Mean age: 52 -Gender: 4M/20F EG2 (COVID-19-severe/critical) (n.26) -Mean age: 66 -Gender: 18M/8F CG: N/A	Primary: 6MWT. Secondary: -Complete effort test (only in the subgroup of patients with severe/critical COVID-19); Shuttle walking test (ESWT); Maximum isometric knee extension force (MicroFET 2 dynamometer) and grip force (Jamar manual dynamometer) evaluated by dynamometry; Test sit-to-stand; FVC; FEV _1_), TLC; DLCO; MRC; SF-36; GAD-7; PHQ-9	At baseline and at the end of the PR program	Measures of FVC or FEV1 lung function improved significantly in the range of 7.7–15.7% in both groups. Quality of life improved significantly only in patients with severe/critical COVID-19 in the SF-36 mental sum score (38.5 to 52.9 points; *p* < 0.001).
Hameed et al., 2021 [28]	Prospective cohort study	EG1 (MTM): virtual physical therapy via a telemedicine platform. EG2 (HPT): Home Physical Therapy EG3 (IE): independent exercise program CG: No therapy.	1–2 times a week with sessions of 30–60 min.	EG1 (VPT) (n.44) -Mean age: 60 -Gender: 53%M/57%F EG2 (HPT) (n.25) -Mean age: 57 -Gender: 86%M/24%F EG3 (IE) (n.17) -Mean age: 59 -Gender: 65%M/35%F CG (n.20) -Mean age: 58 -Gender: 55%M/55%F	Primary: Change in lower limb strength; sit-to-stand test; 2MWT. Secondary: N/A	At baseline and 2-week follow-up.	At follow-up, 65% of patients in the VPT group and 88% of patients in the HPT group achieved clinically significant difference for improvement in sit-to-stand scores, compared with 50% and 17% of those in the IE group and the non-exercise group (*p* = 0.056). The clinically significant difference for step test improvement was met by 74% of patients in the VPT group and 50% of patients in the HPT, IE and non-exercise groups (*p* = 0.12).
Hayden et al., 2021 [29]	Prospective observational study	EG1 (Severe Acute COVID-19); EG2 (Severe COVID-19); EG3 (Mild COVID-19): the program has been adapted to individual needs. Physical training with resistance and strength training, vibratory training for the whole body and inspiratory muscle training. Respiratory physiotherapy with individual training on breathing, seminar on coughing techniques and mucolytic inhalation therapies. General physiotherapy with mobility and gait training. Psychosocial support, nutritional counselling and occupational therapy. CG: N/A	3 weeks Average duration of PR treatment: 26.3 ± 5.9 days	EG1 (Severe Acute) (n.55) -Mean age: 57.9 ± 10.8 -Gender: 34M/21F EG2 (Severe) (n.32) -Mean age: 54.0 ± 9.9 -Gender: 21M/11F EG3 (Mild) (n.21) -Mean age: 52.1 ± 6.8 -Gender: 4M/17F CG: N/A	Primary: NRS; MRC Secondary:Cardinal symptom: list of symptoms associated with COVID-19; 6MWT, FEV1, vital capacity (CV), residual volume (VR), total lung capacity (TLC), total specific airway resistance (sRtot), maximum inspiratory pressure (PImax) and capillary carbon monoxide alveolus diffusion (TLCO) Blood gas analysis, Laboratory blood test, NRS, BFI, EQ-5D-5L, PHQ-9, GAD-7, GROC, Estimation of the overall effectiveness of rehabilitation from the point of view of the patient: 11-point Likert scale	At baseline and at the end of the PR program.	PR was effective after acute COVID-19 in all three groups analyzed. 6MWT improved with large effect sizes in all groups, with major changes in subgroups 1 and 2. Groups 1 and 2 showed statistically significant improvements with moderate to high effect sizes in VC%, TLC%, FEV1%, TLCO_SB% and PImax. Significant decrease in fatigue was observed in groups 1 and 2, with large effect sizes. Anxiety values decreased, with moderate to high effect sizes. All groups showed significant improvement with high effect sizes in QoL.
Hermann et al., 2020 [30]	Cohort study	EG1 (ventilated), EG2 (unventilated): individualized training including aerobic exercise and strength training. Respiratory physiotherapy consisted of teaching breath control (breathing of ruffled lips, mobilization of secretions and diaphragmatic breathing), energy-saving techniques and controlled cough exercises. Twice a week (1 h each), all patients participated in educational sessions. CG: N/A	2–4 weeks with 25–30 therapy sessions, which took place on 5–6 days a week.	EG1 (ventilated) (n. 12) -Mean age: 64.3 -Gender: 9M/12F EG2 (not ventilated) (n. 16) -Mean age: 67.4 -Gender: 5M/16F CG: N/A	Primary: 6-MWT, CRQ, FIM, CIRS, HADS -Patients’ feelings about their actual well-being: Sensitive thermometer (FT) -Lung function, blood gas analysis and oxygen therapy Secondary: N/A	At baseline and at the end of the PR program.	Significant improvements were observed in 6-MWT (+130 m) and FT (+40 points) for the total cohort with no significant differences in intergroup comparison, between ventilated and unventilated patients. Pulmonary function tests showed persistent obstructed ventilation only in a few cases, however, still a part of the patients had limited ventilation and reduced diffusion capacity with the following results: mean FEV1 56%, mean FEV1% FVC 81%, mean TLC 62%, DLCO 56%.
Li et al., 2021 [19]	RCT	EG: via a smartphone application called RehabApp: breathing control and chest expansion, aerobic exercise and LMS exercises specified in a three-level exercise plan with difficulty and intensity programmed to increase over time. CG: Brief Educational Instructions	3–4 sessions per week for 6 weeks.	EG (TERECO) (n. 59) -Mean age: 49.17 -Gender: 26M/34F CG (n.61) -Mean age: 52.03 -Gender: 27M/32F	Primary: 6MWT Secondary: -Squat time in seconds to evaluate the muscle strength of the lower limbs; -Lung function: spirometry, SF-12, mMRC	At baseline, at 6 weeks (post-treatment) and at 28 weeks (follow-up).	The 6MWT in the TERECO group improved by 80.2 m in the after-treatment period, while in the control group there was a small improvement of 17.1 m. Lower limb muscle strength (SML) improved to a greater extent in the TERECO group, treatment effects were 20.12 sec post-treatment and 22.23 sec at follow-up. Lung function parameters improved in both groups, except for maximal voluntary ventilation (MVV) which improved post-treatment most in the TERECO group. The increase in the physical component of SF-12 was greater in the TERECO group with treatment effects estimated at 3.79 post-treatment and 2.69 following.
Liu et al., 2021 [21]	RCT	EG: training of respiratory muscles; exercise for cough; diaphragmatic training; stretching exercise; and exercise at home (ruffled lip breathing and cough training: 30 sets per day). Exercises for upper limb in flexion, horizontal extension, abduction and external rotation. CG: no treatment	2 sessions per week for 6 weeks.	EG (n. 36) -Mean age: 69.4 -Gender: 24M/12F CG (n.36) -Mean age: 68.9 -Gender: 25M/11F	Primary: Respiratory function: forced expiratory volume in 1st second forced expiratory volume(FEV1); Forced vital capacity (CVF); capillary alveolus diffusion of carbon monoxide DLCO (%) Secondary:6MWT, FIM, SF-36, SDS, SAS	At baseline and at the end of the PR program.	In the intervention group, significant differences were found in FEV1(L), FVC(L), FEV1/FVC%, DLCO% and 6MWT. SF-36 scores, in 8 dimensions, were statistically significant within the intervention group and between the two groups. SAS and SDS scores in the intervention group decreased after the intervention, but only anxiety had a significant outcome within and between the two groups.
Pehlivan et al., 2022 [20]	RCT	EG (TeleGr): patient education, rhythm running/autonomous walking in the corridor, breathing exercises, active cycle of breathing techniques, range of motion exercises and standing squats. The exercises were performed 10 times per session. The number of repetitions has been adjusted according to the fatigue rate. CG: exercise brochures with the same content (patient education, breathing exercises, movement exercises, self-walking and squats)	3 days a week for 6 weeks.	EG (TeleGr): (n.34) -Mean age: 50.76 -Gender: 14M/3F CG: (n.34) -Mean age: 43.24 -Gender: 6M/11F	Primary: MRC; VAS, TUG, Short Battery for Physical Performance (SPPB) includes three tasks: a standing balance test (side-by-side, semi-tandem and tandem), a usual gait speed of 4 m and sitting and getting up 5v from a chairSt. George’s breathing questionnaire, BDI Secondary: N/A	At baseline and at the end of the PR program.	Significant improvement in EG (TelerGr) in terms of mMRC (*p* = 0.035), 30STS (*p* = 0.005), 5 sitting to standing time, which is one of the subtests of SPPB (*p* = 0.039) and SGRQ scores. A significant improvement was observed only in the pain score in the CG (*p* = 0.039). Statistically significant difference between groups in SGRQ (*p* = 0.035) and total (*p* = 0.042) scores. In addition, a more symptomatic improvement was found in TeleGr.
Puchner et al., 2021 [31]	Observational cohort study	EG: Respiratory therapy; Training of respiratory muscles; Mobilization and perception of breath; Endurance and strength training; Speech therapy intervention and swallowing evaluation; Occupational therapy, psychological therapy. CG: N/A	25–50 min sessions for 3 weeks.	EG (n.23) -Mean age: 57 ± 10 -Gender: 16M/7F CG: N/A	Primary:Respiratory function by spirometry: forced vital capacity (FVC), forced vital capacity in one second (FEV1), FEV1/FVC, total lung capacity (TLC), residual volume (RL) and diffusion capacity for carbon monoxide (DLCO) and blood gas analysis pH, pO2 and pCO2, MIP, 6MWT, BI	At baseline and at the end of the PR program.	Significant improvement in lung function: increased forced vital capacity (FVC), forced expiratory volume in one second (FEV_1_), total lung capacity (TLC) and carbon monoxide diffusion capacity (DLCO). The state of physical performance has improved: average increase of 176 m in 6MWT and significant improvement in the Barthel Index (BI). Lung function is still impaired in 57% of all patients, and 83% of all study participants still had DLCO reduction at the end of the rehabilitation program.
Spielmanns et al., 2021 [32]	Prospective observational study	EG and CG: resistance training (cycling and treadmill), 3-level gymnastics, 3-level indoor and outdoor walking and strength training; Respiratory physiotherapy: breathing with wrinkled lips, mobilization of secretions and diaphragmatic breathing), energy-saving techniques and controlled cough exercises; Twice a week (1 h each): educational sessions (self-management, coping skills, self-medication, infection and exacerbation management, dyspnea, oxygen use and nutritional interventions).	25–30 therapy sessions in 5–6 weekdays for 3 weeks.	EG (n.99) -Mean age: 67.72 -Gender: 57M/42F CG: (n.419) -Mean age: 69.28 -Gender: 206M/213F	Primary: 6MWT, Pulmonary function test: spirometry and plethysmography and blood gas analysis,CRQ, FIM, HADS, CIRS -Wellness: Sensitive thermometer (FT) Secondary: N/A	At baseline and at the end of the PR program.	Improvements in 6-MWT in pre-post comparison averaged 180 (±101) meters for EG and 102 (±89) meters for CG (*p* < 0.001). FT showed a significant improvement for post-COVID-19 patients (PG) of 21 (±14) points and for patients with other lung disease (LG) of 17 (±16) points (*p* < 0.039), while FIM increased significantly by 11 (±10) points in post-COVID-19 patients (PG) and 7 (±8) points in patients with other lung disease (LG) (*p* < 0.001).
Vitacca et al., 2020 [34]	Practical guideline	/	/	/	/	/	/
Vitacca et al., 2020 [35]	Declaration of consent	/	/	/	/	/	/
Zampogna et al., 2021 [33]	Multicenter retrospective observational study	EG: Level A: SPPB < 6 with a 1:1 physiotherapist/patient ratio with mobilization, active exercises and free walking, peripheral limb muscles, shoulders and upper limb activity. Level B: SPPB ≥ 6 with a 1:4–5 physiotherapist/patient ratio with gymnastics, strengthening, balance exercise, rhythmic walking, and thoracic physiotherapy with bronchial hygiene techniques and lung expansion procedures. CG: N/A	From a minimum of 1 session a day of 20 min up to 2–3 sessions of 30 min a day.	EG (n.140) -Mean age: 71.0 -Gender: 95M/45F CG: N/A	Primary: -Exercise tolerance: 6 Minutes Walking Test (6MWT); -Function of the lower limbs: Short Physical Performance Battery (SPPB); -Motor performance: the Barthel Index (BI). Secondary: N/A	At baseline and at the end of the PR program	Improvements in lower limb function in SPPB and motor performance in BI with scores ranging from 55 to 95. 81 patients after rehabilitation treatment were able to complete the 6 MWT with an average distance of 285 m. The percentage of patients who at the time of admission were unable to stand, get up from a chair and walk after rehabilitation is significantly reduced.

FEV1: forced expiratory volume in the 1st second; CVF: forced vital capacity; PImax (cmH2O): inspiratory pressure; PEmax (cmH2O): aspiratory pressure; 6MWT: six minute walking test; Tinetti scale: balance; Borg test: dyspnea; muscle strength; St George questionnaire: quality of life (QoL); Pichot’s questionnaire: fatigue; hospital anxiety and depression questionnaire: anxiety and depression; post-traumatic stress disorder checklist scale (PCLS): post-traumatic stress; FIM: functional independence measure; CRQ: chronic respiratory questionnaire; Medical Research Council (MRC); time up and go (TUG); activity-specific balance confidence: activity-specific balance confidence scale (ABC); general self-effectiveness: patient-reported outcomes measurement and information system (PROMIS) short form (SF); clinical frailty scale (self-reported) and patient health questionnaire 8 (PHQ8); cognitive assessment: Montreal cognitive assessment (MoCA)-blind; Barthel index (BI); Borg rating of perceived exertion (RPE scale); COPD assessment test (CAT); functional assessment of chronic illness therapy fatigue scale (FACIT); anxiety and depression: hospital anxiety and depression scale (HADS); EuroQual 5 domains (EQ5D); total lung capacity (TLC) and carbon monoxide diffusion capacity (DLCO); quality of life: related to health (SF-36); anxiety symptoms: generalized anxiety disorder-7 (GAD-7) questionnaire; depressive symptoms: patient health questionnaire (PHQ-9).

## Data Availability

Not applicable.

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
