# Peer review of "Effectiveness of Respiratory Rehabilitation in COVID-19’s Post-Acute Phase: A Systematic Review"

_healthcare, 2023, doi:10.3390/healthcare11081071_

Round 1
Reviewer 1 Report
line 12, not everyone is impacted say maybe say instead that it may impact
include a recommendation
line 41 ranging from asymptomatic infection to a slight sickness ...
line 45 say what ARDS is first time used
line 53, new sentence from 'whereas'
line 55 should be 'does not'
line 62 not clear what ample refers to and does not relate to 'varied'
line 65 sentence is not clear. Maybe made 2 sentences. Use 'does not'
line 76 is not clear and sentence is too long. Make 2 sentences
line 92 sentence is too long and unclear
line 117 onwards formatting issue
line 162 should be 'allowing'
line 173 - did not meet the required inclusion
line 213 use gender instead of sex
better to start sentence with words rather then a number - line 221 - 'four'
line 262 is repeated
line 265 is not clear
line 267 did not
line 280 ? used a stress test of 1 minute ..
line 284 start new sentence 'only 3'
line 298 did not
line 380 did not
line 392 not clear - contingency of the topic - not appropriate language
line 394 does not
line 402 not clear
line 413 - constant aspect of what. Sentence too long and not clear
line 445 sentence too long and unclear
line 502 convenient to argue the implementation of - sentence incomplete
recommendatins
This was very hard to read as there was a combination of long sentences that were not clear and short paragraphs and no links between points. A paragraph is more then two sentences.
Needs serious editing and help with English writing
Author Response
CORRECTIONS |
RESULTS |
line 12, not everyone is impacted say maybe say instead that it may impact. Include a recommendation. |
“This condition may impact on the quality of life of survivors and sums up with sizable pulmonary and respiratory outcomes. Respiratory rehabilitation is known for its effects in improving dyspnea, alleviating anxiety and depression, reduce complications, prevent and ameliorate dysfunctions, reduce morbidity, preserve functions and improve subjects’ quality of life as much as possible. For this reason, respiratory rehabilitation may be recommended for this category of patients.” |
line 41 ranging from asymptomatic infection to a slight sickness ... |
“The infection’s clinical spectrum is wide and varied, ranging from asymptomatic infection to a slight sickness of the higher respiratory tract, a moderate illness to a critical one which can be described as a serious viral pneumonia with respiratory distress, septic shock and/or multiple organ failure [2]” |
line 45 say what ARDS is first time used |
“41,8% of infected subjects developed ARDS and 52,4% of these died [3]. The American-European Consensus Conference on ARDS defined the acute respiratory distress syndrome (ARDS) as a process of nonhydrostatic pulmonary edema and hypoxemia associated with a variety of etiologies and carries a high morbidity, mortality (10to 90%) [4]”.
|
line 53, new sentence from 'whereas'
|
This paragraph has been revised and deleted from the text. |
line 55 should be 'does not'
|
“The issue for the survivors of COVID-19 does not terminate with the end of the pulmonary inflammation since a significant number of patients continue signaling persistent symptoms way beyond the acute phase of the sickness.” |
line 62 not clear what ample refers to and does not relate to 'varied'
|
“ These medium and long-term effects, known as post-COVID-19 Syndrome, signs and symptoms that continue for more than 12 weeks or “Long Covid”, which includes both ongoing symptomatic COVID‑19 (from 4 to 12 weeks) and post‑COVID‑19 syndrome (12 weeks or more), are extremely varied and extensive.” |
line 65 sentence is not clear. Maybe made 2 sentences. Use 'does not'
|
“ Respiratory damage plays a crucial role within patients who have passed COVID-19, since the removal of the cause of lung damage does not hinder the development of fibrotic and progressive in-terstitial lung disease.” |
line 76 is not clear and sentence is too long. Make 2 sentences
|
“Rehabilitation programmes play a crucial role in combating the pandemic, in addition to the use of vaccines [12], as they are an effective means of containing the adverse effects of COVID-19 on general public health [13].”
|
line 92 sentence is too long and unclear
|
“The objective of this study is to conduct a systematic review of scientific literature to assess the efficacy and benefits of pulmonary rehabilitation (PR) programs in the post-acute phase of COVID-19, useful to promote an improvement of respiratory functions, autonomy and quality of life (QoL) of people affected by COVID-19 and reduce the incidence and severity of lung complications.”
|
line 117 onwards formatting issue
|
We changed the formatting at the indicated point. |
line 162 should be 'allowing'
|
“The randomized controlled trials (RCT) included were evaluated with the use of the Physiotherapy Evidence-Based Database (PEDro) scale, which is considered a reliable instrument allowing critical evaluation of methodologic quality in experimental studies on physical therapy.” |
line 173 - did not meet the required inclusion |
“The practical guidelines and declaration of consent have been included, as they are relevant for the revision, but not methodologically evaluated.”
|
line 213 use gender instead of sex
|
“Gender of patients is prevalently male. The data related to the overall population included in the analysed studies is shown in Table 5.”
|
Better to start sentence with words rather then a number - line 221 - 'four' |
“Four studies have adopted a respiratory physiotherapy program which comprises of deep breathing, inhaling muscle training and bronchial hygiene techniques [23,29,19,33].” |
line 262 is repeated |
We don’t modify the text because this phrase is not repeated, likely misunderstanding, means patients with other pneumonia and patients with other types of lung disease. |
line 265 is not clear + line 267 did not |
“3 studies included a control group, which received educational sessions [18,21,31]. Instead, two studies included a control group which received no treatment [20,26]. Lastly, 4 of the included studies did not exhibit any control group [24, 25,19,32] (Table 5).”
|
Line 280 used a stress test of 1 minute |
It’s an error that we have corrected.
“2 studies also use as stress test the 30 second Sit to Stand [26,28]”
|
line 284 start new sentence 'only 3'
|
“Only 3 studies evaluated autonomy in the activities of daily life (ADL) through the Barthel Index (BI) [24,19,33].”
|
line 298 did not
|
“9 studies did not include the evaluation of pulmonary function as an outcome [18,31,23,26,27,28,33].” |
line 380 did not
|
“Lastly only 1 study didn’t exhibit any significant differences in pulmonary function between the two groups that underwent the same rehabilitation program but were distinguished in ventilated and did not” |
line 392 not clear - contingency of the topic - not appropriate language. + Line 394 does not
|
“ However, the low number of high-quality studies present in literature does not allow the recognition of an elective protocol.” |
line 402 not clear
|
“Therefore, the studies we analysed in this review show results related to 2 macro-areas: lung function and exercise capacity, evaluated by the Six Minute Walking Test (6MWT).”
|
line 413 - constant aspect of what. Sentence too long and not clear
|
“Respiratory physiotherapy is associated with strength and endurance training of the upper and lower limbs, aerobic exercise, muscle relaxation techniques, and in one case balance and coordination exercises [24]. In order to achieve progressive functional improvement and a reduction in the level of disability.”
|
line 445 sentence too long and unclear
|
“ In the period of the pandemic there was a high percentage of patients with damage and limitations of physical and lung function, remote rehabilitation has proved to be a viable alternative that could become a new frontier in rehabilitation, to ensure greater continuity in patient care.”
|
line 502 convenient to argue the implementation of - sentence incomplete, recommendatins
|
Therefore, it is advisable to increase the implementation of comprehensive and customized rehabilitation protocols for Covid-patients19, in which a large part is devoted to respiratory rehabilitation to allow a recovery of overall function and an improvement in the quality of life. |
Reviewer 2 Report
The authors present a systematic review regarding the effectiveness of rehabilitation programs in COVID-19. There are major issues that are raised, and I am concerned about them. Extensive editing of English language is required.
Major comments:
1. I suggest the introduction to be re-organized. It is not well structured. I think that the introduction does not permit readers to be introduced in the subject. For example, in the 4th and 5th paragraph you refer to long-COVID, but then you change presenting fibrosis in the context of ARDS. Please try to be clearer. Also, fibrosis in post COVID is under investigation. Is there lack of evidence or guidelines regarding suggestion in rehabilitation programs in COVID-19 patients?
2. Regarding methods, I would like to ask why you exclude serious COVID-19 cases from your search, as in these patients, rehabilitation may be more necessary to recover.
3. Please in the supplementary material, include a table with the search terms you used and the associated results in each database you used.
4. You could use a Table for the PICO strategy.
5. 3.5 outcome: please rephrase the 1st line to be better understood by readers.
6. You could insert a Table with the studied outcomes and the results of the studies you finally included in your results. It would be better presented on this way.
7. In the results section, you just state the studied programs or the number of studies you reviewed but you don’t achieve to organise and present the data regarding the effectiveness of the rehabilitation programs you reviewed.
8. What was the impact of these studies on the functional capacity, 6MWT or quality of life of the participants?
9. Regarding telemedicine programs, what was the differences in the involved exercises? What was the impact and the effectiveness of these programs on patients’ recovery?
10. The same I would suggest regarding discussion. You don’t achieve to address the question about effectiveness of the rehabilitation programs. You only state some results, please organise them in order to permit to readers to formulate safe conclusions.
11. What was the impact of diverse rehabilitation programs on functional tests, on quality of life? Did long term follow up was included in the studies you reviewed?
12. Regarding hospitalised non-ICU COVID-19 patients, what were the conclusions? What is the impact of these programs on their recovery?
Minor comments:
1. Please replace in the manuscript COVID-19 and SARS-CoV-2 instead of covid-19 and Sars-Cov-2.
2. In the abstract and in the 1st paragraph of discussion please replace efficacy with effectiveness, as you investigated in real life the impact of rehabilitation programs.
3. Line 39 in introduction, please rephrase it ‘… and with that ‘ to ‘6.618.579 deaths have been recorded so far’.
4. Line 354 in Discussion. Please rephrase nothing improvements.
Author Response
MAJOR COMMENTS |
RESULTS |
. 1. I suggest the introduction to be re-organized. It is not well structured. I think that the introduction does not permit readers to be introduced in the subject. For example, in the 4th and 5th paragraph you refer to long-COVID, but then you change presenting fibrosis in the context of ARDS. Please try to be clearer. Also, fibrosis in post COVID is under investigation. Is there lack of evidence or guidelines regarding suggestion in rehabilitation programs in COVID-19 patients? |
We have reorganized the introduction in a more clear and linear way to allow readers to understand the topic. We, also, mentioned in the introduction the presence of guidelines that suggest rehabilitation programs for COVID-19 patients. |
2. Regarding methods, I would like to ask why you exclude serious COVID-19 cases from your search, as in these patients, rehabilitation may be more necessary to recover |
Were excluded patients who presented Covid in the acute phase because clinically unstable and therefore not subject to respiratory rehabilitation as recommended by the published guidelines. Also, in the acute phase of covid it was not recommended to perform respiratory rehabilitation. |
3. Please in the supplementary material, include a table with the search terms you used and the associated results in each database you used |
In the supplementary material, we put a table with the search terms you used and the associated results in each database you used. See Table 6
|
4. You could use a Table for the PICO strategy. |
We have reported the PICO strategy both in extended form and in a summary table. See Table 1 |
5. 5. 3.5 outcome: please rephrase the 1st line to be better understood by readers.
|
The analyzed studies show main results related to 2 macroareas: lung function and functional capacity (exercise), which were evaluated in almost all studies using different scales and other tools.
|
6. You could insert a Table with the studied outcomes and the results of the studies you finally included in your results. It would be better presented on this way. Table 4 |
We have produced a table summarising the main features of the studies included in the review. See table 5. |
7. In the results section, you just state the studied programs or the number of studies you reviewed but you don’t achieve to organise and present the data regarding the effectiveness of the rehabilitation programs you reviewed.
|
In the section "Summary of results" are analyzed the various rehabilitation programs described in the studies and their effectiveness. |
8. What was the impact of these studies on the functional capacity, 6MWT or quality of life of the participants?
|
We have added an explanation in the section "summary of results" in which the impact of proposed rehabilitation programs on functional capacity, 6MWT and quality of life is explained.
“In the analyzed studies it was observed that Covid-19 patients underwent a complete respiratory rehabilitation program with breathing, aerobic, strength and endurance exercises, carried out both in hospital and remote with telemedicine, have recorded a gradual recovery of functional capacity, operating capacity that translates into an initial increase in the distance traveled to 6MWT. This results in a gradual increase in autonomy and improvement of the quality of life. Some studies have shown that the same rehabilitation program focused on respiratory function could produce an important improvement in functional capacity even in patients with different stages of Covid-19.”
|
9. Regarding telemedicine programs, what was the differences in the involved exercises? What was the impact and the effectiveness of these programs on patients’ recovery?
|
We have added a brief summary of the rehabilitation programmes proposed in the studies analysing telerehabilitation and their differences. You can find it in the "intervention" of results section.
“2 of these studies (Hameed et al and Pehlivan et al) describe a rehabilitation programme delivered on a telerehabilitation platform. In Hameed et al [28], diaphragmatic breathing exercises, Spirometry Incentive, Sit to Stand, Standing Gear, Shoulder Scaption, Standing Heel Lifts, Sidestepping and Wall Flexion are proposed. In Pehlivan et al [20], patient education, rhythm run/autonomous walk in the corridor, breathing exercises, active cycle of breathing techniques are proposed, range of motion exercises and standing squats. Instead in 2 of these studies telerehabilitation program is proposed through a smartphone app. In Capin et al [18] the program includes breathing and compensation techniques, high intensity strength training, aerobic and cardiovascular exercises, exercise balance, functional activities, stretching and lifestyle coaching and motivational talk. In addition, this application is used for facilitate self-directed intervention outside supervised sessions. In Li et al [19] is described the use of an application for smartphones called “RehabApp” and includes chest breathing and expansion, aerobic exercise in a three-level exercise plan with programmed difficulty and intensity to increase over time.”
|
10. The same I would suggest regarding discussion. You don’t achieve to address the question about effectiveness of the rehabilitation programs. You only state some results, please organise them in order to permit to readers to formulate safe conclusions.
|
We reorganized the discussion by focusing on the effectiveness produced by the various rehabilitation programs described in the studies included.
|
11. What was the impact of diverse rehabilitation programs on functional tests, on quality of life? Did long term follow up was included in the studies you reviewed?
|
We reported in the discussion the impact that different rehabilitation programs have on functional tests. One of the important limitations of the included studies is the absence of a long-term follow up to better section understand the long-term benefits of the proposed rehabilitation protocols. We have reported this in the “limits of the review”.
|
12. Regarding hospitalised non-ICU COVID-19 patients, what were the conclusions? What is the impact of these programs on their recovery?
|
We reported in the discussion a brief explanation about the effects of respiratory rehabilitation on non ICU-COVID-19 patients. Hence, we can state that even though we have the presence in scientific literature, concerning Covid’s post-acute phase, heterogeneous pulmonary rehabilitation protocols in terms of suggested exercises, we can still see that a rehabilitation protocol that combines respiratory physiotherapy, aerobic training and strength, without neglecting the neuropsychological aspect produces a positive effect on exercise capacity, lung function, dyspnoea, fatigue, reduction of anxiety and depression and improvement of quality of life (QoL). |
MINOR COMMENTS |
RESULTS |
1. Please replace in the manuscript COVID-19 and SARS-CoV-2 instead of covid-19 and Sars-Cov-2 |
We have replaced in the manuscript COVID-19 and SARS-CoV-2 instead of covid-19 and Sars-Cov-2.
|
2. In the abstract and in the 1st paragraph of discussion please replace efficacy with effectiveness, as you investigated in real life the impact of rehabilitation programs. |
We have replaced efficacy with effectiveness in the abstract and in 1st paragraph of discussion. |
3. Line 39 in introduction, please rephrase it ‘… and with that ‘ to ‘6.618.579 deaths have been recorded so far’.
|
We have rephrased the sentence in line 39 as indicated. “Since the beginning of the pandemic 640.395.651 confirmed cases worldwide have been registered and 6.618.579 deaths have been recorded so far..” |
4. Line 354 in Discussion. Please rephrase nothing improvements.
|
“4 studies analysed a single cohort of Covid patients without a control group and have noted a significant improvement in functional capacity in terms of increase in the 6MWT, the ISWT, the ESWT and the SPPB, as well as an improvement in autonomy in daily life activities (ADL) assessed through the Barthel index. [19,24,25,33].” |
Reviewer 3 Report
The purpose of this study is to conduct a systematic review of the scientific literature to assess the effectiveness and benefits associated with the introduction of pulmonary rehabilitation programs in the post-acute phase of Covid-19. The authors substantiate that the combination of various exercises, such as aerobic, respiratory, fitness and strength, without taking into account the neuropsychological part, proved effective for improving health, well-being and quality of life in patients after acute Covid. The included studies show that a remote rehabilitation program deployed through a telemedicine platform is effective. Thus, the topic of the publication is relevant, and its conclusions can be useful as a methodology for the rehabilitation of patients who have undergone COVID-19.
Comments and suggestions:
1. The authors use the term "post-acute phase". However, they do not specify this definition. At the same time, this concept is not standardized in the scientific literature. Most recently, the World Health Organization has provided a definition after COVID-19 as the persistence of symptoms after 3 months of SARS-CoV-2 infection lasting at least 2 months and not explained by any other disease. However, there are other temporary interpretations of this condition. It is also unclear from the text of the article the relation of "post-acute phase" to other related definitions: Post-acute COVID-19 syndrome, long COVID and others. It is also not clear what criteria of "post-acute phase" were used by the authors of the original studies, which were systematized in the presented article.
2. It should be taken into account that the overall NOS score may be higher in the reviewers' assessment compared to the assessments of the authors of the systematic review (doi:10.1186/1471-2288-14-45.)
3. The NOS School was developed to assess the methodological quality of non-randomized studies. At the same time, the initial version of the NOS scale does not specify how many points a study should score to determine its methodological quality as high, medium or low. In one of the Cochrane systematic Reviews, studies that scored less than 6 points on the NOS scale were assigned low methodological quality (doi:10.1002/14651858.CD009849). Therefore, it is advisable to allocate work with < 6 points to a separate group with questionable results.
Also, a study using the PEDro scale is considered to be of medium to high quality if it is rated at least 6-10 points.
4. (119-124) In accordance with the PRISMA principles, the inclusion and exclusion criteria were formulated by defining the PICO strategy. At the same time, P (population) is given as an exception, and in other cases it is not clear where inclusion is and where exclusion is.
Author Response
CORRECTIONS |
RESULTS |
1. The authors use the term "post-acute phase". However, they do not specify this definition. At the same time, this concept is not standardized in the scientific literature. Most recently, the World Health Organization has provided a definition after COVID-19 as the persistence of symptoms after 3 months of SARS-CoV-2 infection lasting at least 2 months and not explained by any other disease. However, there are other temporary interpretations of this condition. It is also unclear from the text of the article the relation of "post-acute phase" to other related definitions: Post-acute COVID-19 syndrome, long COVID and others. It is also not clear what criteria of "post-acute phase" were used by the authors of the original studies, which were systematized in the presented article. |
In the introduction we reported the definition of post-covid syndrome, long covid. But there is no single definition of post-acute phase in the literature, so we have reported what we mean by post-acute phase. We reported this aspect as one of the most important limits of the revision.
“These medium and long-term effects, known as post-COVID-19 Syndrome, signs and symptoms that continue for more than 12 weeks or “Long Covid”, which includes both ongoing symptomatic COVID‑19 (from 4 to 12 weeks) and post‑COVID‑19 syndrome (12 weeks or more), are extremely varied and extensive”.
“In the literature there is no unique and officially recognized definition of the post-acute phase, some authors use this expression to indicate the immediate phase after the acute one (after 4 weeks) with persistent symptoms.” |
2. It should be taken into account that the overall NOS score may be higher in the reviewers' assessment compared to the assessments of the authors of the systematic review (doi:10.1186/1471-2288-14-45.)
3. The NOS School was developed to assess the methodological quality of non-randomized studies. At the same time, the initial version of the NOS scale does not specify how many points a study should score to determine its methodological quality as high, medium or low. In one of the Cochrane systematic Reviews, studies that scored less than 6 points on the NOS scale were assigned low methodological quality (doi:10.1002/14651858.CD009849). Therefore, it is advisable to allocate work with < 6 points to a separate group with questionable results. Also, a study using the PEDro scale is considered to be of medium to high quality if it is rated at least 6-10 points. |
We have divided the studies analyzed with the Scala NOS and Pedro Scale according to their methodological quality based on what you have recommended. You can find in the section “Results”.
“In general, a study with a score of at least 6-10 points is considered to be of medium-high quality. The final score obtained in the evaluation of the randomized controlled trials included varies from 5 to 7, with an average of 6. In particular 3 of the included RCTs are of medium-high quality [18, 19 and 20], only one RCT is of medium quality [21].”
“In particular, 5 cohort studies [25, 29, 30,31,32] and 2 observational studies [24 and 33] obtained less than 6 points on the NOS scale and were given low methodological quality. In this case the results of these studies should be considered as questionable compared to the others.” |
4. (119-124) In accordance with the PRISMA principles, the inclusion and exclusion criteria were formulated by defining the PICO strategy. At the same time, P (population) is given as an exception, and in other cases it is not clear where inclusion is and where exclusion is. |
We have reformulated the PICO strategy so that the inclusion and exclusion criteria are made clearer and more explicit.
“P (population): patients diagnosed with COVID-19 in post-acute phase and clinically stable were included. Instead, serious COVID-19 cases or acute-phase cases with clinical instability were excluded. I (intervention): were included respiratory physiotherapy in its different means, either delivered in conventional form (in person) or through telemedicine. Other forms of rehabilitation have been excluded. C (comparison): patients that only receive standard assistance/cure or receive no cure whatsoever. O (outcome): improvement of the respiratory function and physical performance, reduction of dyspnoea, fatigue and improvement of autonomy and quality of life in patients affected by COVID-19. “
|
Reviewer 4 Report
If you follow the PRISMA guideline you need to review the steps and the methodology. Make clear the inclusion/exclusion criteria.
You need to detail the reasons and the number of articles for every reason in the PRISMA flow.
You evaluated the quality. What about the risk of bias?
You need to present the characteristics of the included studies in a new table.
Where is Table 4?
Author Response
CORRECTIONS |
RESULTS |
1.If you follow the PRISMA guideline you need to review the steps and the methodology. Make clear the inclusion/exclusion criteria.
|
We have reformulated the PICO strategy so that the inclusion and exclusion criteria are made clearer and more explicit. We also tried to change the paragraph names of the article to comply with the prism checklist. |
2. You need to detail the reasons and the number of articles for every reason in the PRISMA flow. |
In the PRISMA flow chart we reported the number of the records included and excluded for each stage of the selection and the reason why they were excluded.
|
4. You need to present the characteristics of the included studies in a new table. |
The main features of the study included in the review have been reported in the Table 5. |
5. Where is Table 4? |
You can find the Table 4, which is now called Table 5, at the end of the manuscript.
|
Round 2
Reviewer 1 Report
Thanks you for the opportunity to check these changes and generally they are good. My only concern is that lines 245, 250, 252, 271, 288, 292, 296 - should not start sentence with number (3), should write out (three).
Also the last three lines need a link between the points rather then starting consecutive paragraphs with the same number - for instance - another, in addition.
Author Response
All required changes have been made
Best Regards
Giovanni Galeoto
Reviewer 4 Report
I do not think the limits of the review must be in a separate section. Please include them into the Discussion section.
The first column of the tables 2-5 must be the study name written as first author' name and the year of publication followed by the reference's number, as for example Capin 2022 [18]
Author Response

(The authors gave the same response as above.)
